# The lead and cadmium content in rice and risk to human health in China: A systematic review and meta-analysis

Xianliang Huang[1,2], Bo Zhao[1,2], Yanlei Wu[1,2], Mingtian Tan[1,2], Lisha Shen[3], Guirong Feng[1,2], Xiaoshan Yang[1], Shiqi Chen[1,2], Youming Xiong[1], En Zhang[1], Hongyu Zhou[4,5]*

1 Food Laboratory, Chongqing Institute for Food and Drug Control, Chongqing, China, 2 Key Laboratory of Condiment Supervision Technology for State Market Regulation, Chongqing, China, 3 Chongqing Academy of Chinese Materia Medica, Chongqing, China, 4 College of Public Health and Management, Chongqing Medical University, Chongqing, China, 5 The First Affiliated Hospital of Chongqing Medical University, Chongqing, China

* hyzhou@cqmu.edu.cn

**Data Availability Statement:** All relevant data are within the paper and its Supporting Information files.

**Funding:** (1) Recipient: X.S.Y. Grant number: 2017YFC1602000, Funding Source: the National

## Abstract

Numerous studies have investigated concentrations of lead (Pb) and cadmium (Cd) in rice in China, but have come to divergent conclusions. Therefore we systematically reviewed and meta-analyzed the available evidence on levels of Pb and Cd in rice in different regions of China in order to assess the potential risk to human health. The meta-analysis included 24 studies of Pb levels and 29 studies of Cd levels, published in 2011–2021. The pooled Pb concentration in rice was 0.10 mg per kg dry weight (95% CI 0.08−0.11), while the pooled Cd concentration was 0.16 mg per kg dry weight (95% CI 0.14−0.18). These levels are within the limits specified by national food safety standards. However, the total target hazard quotient for both metals exceeded 1.0 for adults and children, suggesting that rice consumption poses a health risk.

## Introduction

Rice is a well-known staple food, consumed by about 50% of the population in more than 100 countries around the world. As the most populous country in the world, China is the largest producer and consumer of rice in the world. China's annual rice production totals approximately $2.07 \times 10^{11}$ kg and accounts for nearly 34% of total global output [1–3]. Contamination of heavy metals is mainly caused by natural origination and anthropogenic activities, of which the latter one (include industries of mining, fertilizers, and pesticides) made predominant contribution, have led to the continuing accumulation of toxic heavy metals in the soil of rice paddies, from which the metals can enter rice [1, 4–6]. This accumulation is especially high in southern China, which has rapidly industrialized [7].

Many studies have shown that the heavy metal content in rice exceeds food safety standards in China [8], especially levels of cadium (Cd) and lead (Pb) [9–11]. The legal limit for both metals in rice is 0.2 mg/kg in China. The mean Cd levels in rice grain have been reported to be

Key R&D Program of China. (2) Recipient: X.L.H. Grant number: cstc2021jxjl130009, Funding Source: Chongqing Performance Incentive Guidance Special Project of Chongqing Science and Technology Bureau. (3) Recipient: X.S.Y. Grant number: cstc2018jscx-mszdX0122, Funding Source: the Key demonstration project of Chongqing Technology Innovation and application demonstration project of Chongqing Science and Technology Bureau. The funders had no role in study design, data collection and analysis, decision to publish, or preparation of the manuscript.

**Competing interests:** The authors have declared that no competing interests exist.

0.69 mg/kg in Xiangtan County of Hunan Province [12], 0.62 mg/kg in Shaoguan City of Guangdong Province [13], and 0.29 mg/kg along the Yangtze River in Hubei, Hunan, and Jiangxi Provinces [14]. The study in the Yangtze River area has also reported a mean Pb level of 0.25 mg/kg in rice grain [14].

Elevated dietary consumption of Pb and Cd from rice may harm human health [15, 16]. Cd can damage kidneys as well as the pulmonary, cardiovascular, and musculoskeletal systems. Elevated Cd consumption has also been linked to Itai-Itai symptom [17–20]. Pb, for its part, can damage the immune, digestive, and nervous systems, as well as compromise cognitive development [21–23]. Several studies in different regions of China have assessed whether levels of Pb and Cd in rice pose a health risk [24–26], but they have come to divergent conclusions. For example, a study in Guizhou Province concluded that levels of Cd and Pb in rice were too low to pose a health risk [27], while a study in the Pearl River Delta concluded the opposite [28]. The relatively small samples in individual studies has prevented a coherent, overall evaluation of risk.

Therefore we aimed to 1) investigate, even via a meta-analysis of the existing literature, the presence of Pb and Cd in rice from many areas in China; and 2) assess the potential human health risks associated with long-term exposure.

## Materials and methods

### Search strategy

Two authors (B.Z. and G.R.F.) searched for relevant studies in PubMed, Web of Science and ScienceDirect databases that were published from January 2011 through October 2021. The search string was "rice" AND ("heavy metal" OR "lead" OR "cadmium") AND "China". Only studies published in English were considered. Reference lists in selected articles and relevant review articles were manually searched to identify additional studies.

### Inclusion and exclusion criteria

After the initial screening, the full text of potentially eligible articles were downloaded and evaluated carefully according to the inclusion and exclusion criteria. The studies were included if they measured levels of Pb and Cd in rice in China, were published in English, and were available as full text. Studies were excluded if they measured metal levels in cooked rice, rice planted on an experimental farm, rice paddies located near mining and smelting areas, or rice samples collected from markets.

### Definitions and data extraction

Two authors (M.T.T. and L.S.S.) independently evaluated and extracted data from the included studies using a predefined, standardized protocol. The extracted data on general characteristics of studies included the first author, year of publication, years of sampling, journal of publication, sample size, study area, assay method, average concentration and standard deviation (SD). One study [29] reported ranges, which we converted to SD as described (When the sample size between 25 and 70, Range/4 is the best estimator for the standard deviation) [30]. Disagreements about extracted data were resolved through discussion.

### Quality assessment

Two authors (X.L.H. and Y.L.W.) independently evaluated the quality of included studies using the Combie evaluation tool [31]. Included studies were graded in 7 aspects according to the Combie evaluation tool which is as follows: the study design was scientific and rigorous;

the data collection method was reasonable; the response rate of participants was reported; the total representativeness of samples were favorable; the research objective and methods were reasonable; the power of the test was reported; the statistical method was correct. "Yes", "no" and "have no idea" were respectively utilized to evaluate each item, which was successively given 1 point, 0 points, and 0.5 points. The total score was 7.0 points (6.0~7.0 points, 4.0~5.5 points, and 0~4.0 points were considered to high, medium and low quality respectively) [31]. Differences were resolved through discussion.

## Statistical analysis and meta-analysis

Meta-analysis was performed using STATA 15.0 software (Stata Corp, College Station, TX, USA). Pooled concentrations and 95% confidence intervals (CIs) were calculated for all outcomes. Statistical heterogeneity among studies was assessed based on $I^2$, with 25% defined as low heterogeneity; 50%, moderate heterogeneity; and 75%, high heterogeneity [32, 33]. Meta-analysis was performed using a random-effects model if $I^2 > 50\%$ [34]; otherwise, a fixed-effect model was used. Meta-regression was used to identify studies that might explain the observed heterogeneity; the covariates in this regression were years of sampling, study area, assay method, sample size, and quality score. Sources of heterogeneity were also explored through meta-analysis of subgroups defined by years of sampling, study area, assay method, sample size and quality score.

Sensitivity analysis was conducted by omitting studies one by one, and the P values of pooled concentrations were compared. The results were considered robust if the P values were not substantially different. Publication bias was quantitatively analyzed using Egger's test [35], and risk of bias was considered significant if $P < 0.05$.

## Health risk assessment

The target hazard quotient (THQ) developed by the US Environmental Protection Agency [36] was used to assess the potential human health risks associated with long-term exposure to heavy metal pollutants in rice. The THQ was calculated as

$$\text{THQ} = \frac{E_F \times E_D \times F_{IR} \times C}{R_{fD} \times W_{AB} \times T_A} \tag{1}$$

where $E_F$ is the exposure frequency per year (365 days); $E_D$, the exposure duration (70 years); $F_{IR}$, the average daily rice intake in kg person$^{-1}$ day$^{-1}$ (0.389 for adults, 0.198 for children) [28, 37]; C, the heavy metal content in rice in mg kg$^{-1}$; $R_{fD}$, the oral reference dose for heavy metals in mg kg$^{-1}$ day$^{-1}$ recommended by the US Environmental Protection Agency (0.001 for Cd, 0.0035 for Pb) [36]; $W_{AB}$, the mean body weight in China in kg person$^{-1}$ (55.9 for adults, 32.7 for children) [28, 37]; and $T_A$, the average exposure time (365 days year$^{-1}$ × 70 years).

Total THQ was calculated as

$$\text{TTHQ} = \sum \text{THQ} \tag{2}$$

across all heavy metal pollutants, which in this study were Pb and Cd. THQ / TTHQ < 1 indicated that the food was safe for human consumption [36].

## Results and discussion

**Study selection.** A total of 2130 articles were retrieved from PubMed, Web of Science, and ScienceDirect databases, and 1561 duplicate articles were excluded. After screening titles and abstracts, we excluded another 327 articles. After carefully reading the full text of the

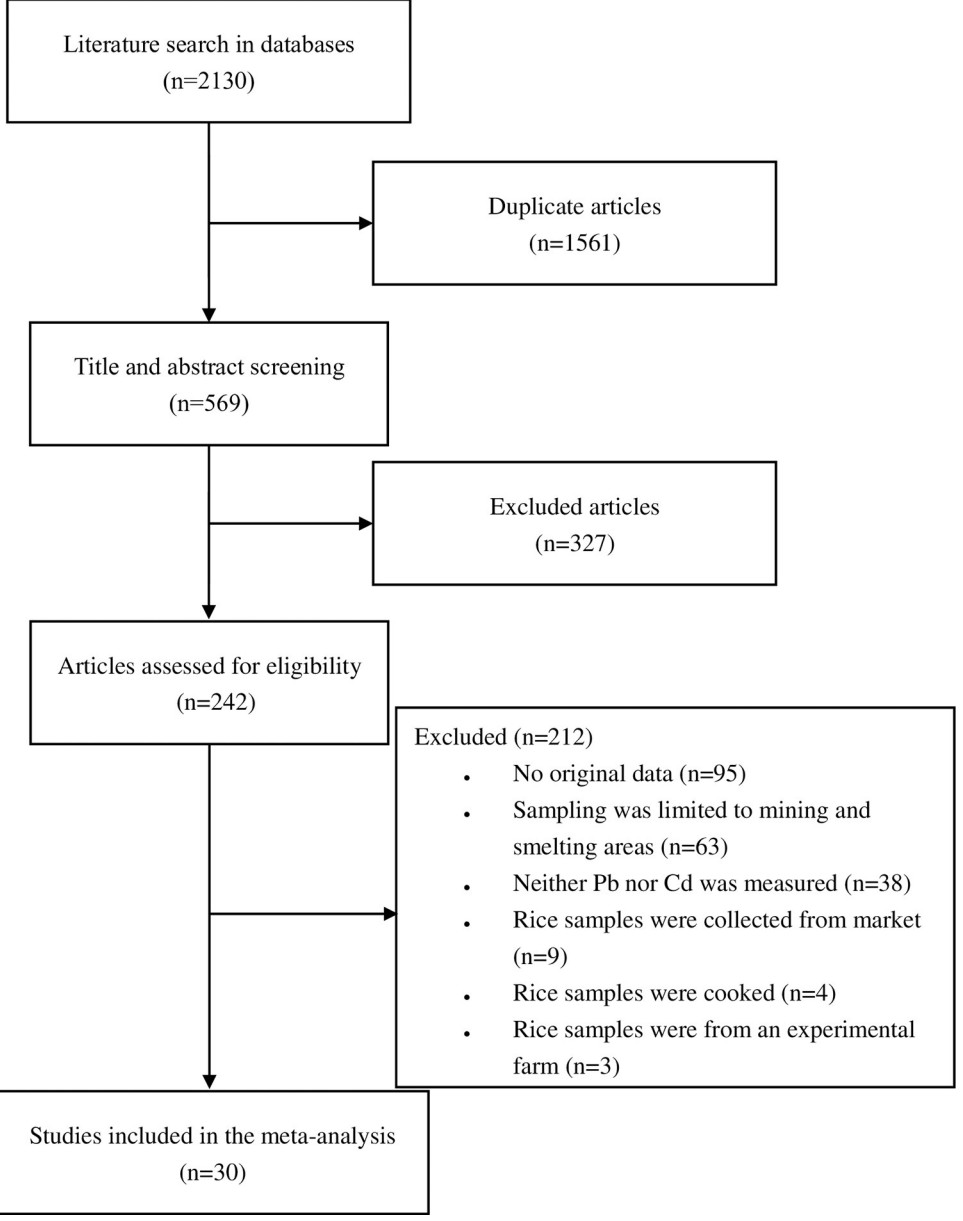

**Fig 1. Flow diagram of study inclusion in the meta-analysis.**

remaining 242 articles, 212 were excluded. Finally, 30 articles were included in the analysis (Fig 1).

## Study characteristics

The main characteristics of the 30 studies are presented in Table 1. The studies were published from January 2011 to October 2021, and they involved a total of 6390 rice samples collected from several major rice-producing areas in China. Among the 30 studies, 24 measured Pb in a total of 5440 rice samples, while 29 studies measured Cd in a total of 6359 rice samples. Concentrations of Pb were determined by inductively coupled plasma-mass spectrometry (ICP-MS, 10 studies), inductively coupled plasma optical emission spectrometry (ICP-OES, 3

**Table 1. Main characteristics of studies included in the meta-analysis.**

| No. | Study | Year(s) of sampling | Area | Sample size | Level (mg/kg dry weight), mean±SD | | Assay method | Quality (Combie points) |
|-----|-------|---------------------|------|-------------|-----------------------------------|---|--------------|-------------------------|
| | | | | | Pb | Cd | | |
| 1 | Zhao et al., 2011 | 2006 | Zhejiang (Wenling) | 96 | NR | 0.072 ±0.105 | GFAAS | Medium (5.5) |
| 2 | Hu et al., 2013 | 2009–2011 | Northeast/Northern China/Northwest/Eastern China/ Central China/Southern China/Southwest | 92 | 0.10±0.14 | 0.08±0.07 | GFAAS | High (6.5) |
| 3 | Li et al., 2014 | 2011 | Zhejiang (Wenling) | 219 | NR | 0.132±0.24 | GFAAS | High (6.5) |
| 4 | Mao et al., 2019 | 2011 | Yangtze River Delta (Jiangsu, Zhejiang, Shanghai) | 137 | 0.098 ±0.003 | 0.064 ±0.008 | ICP-MS | High (6.5) |
| 5 | Liu et al., 2016 | 2012 | Yangtze River Region (Hubei, Hunan, Jiangxi) | 101 | 0.25±0.11 | 0.29±0.39 | GFAAS | High (6.0) |
| 6 | Xie et al., 2017 | 2012–2013 | 18 provinces | 110 | 0.0435 ±0.0755 | 0.0650 ±0.1266 | GFAAS | High (6.5) |
| 7 | Gao et al., 2016 | 2013 | Zhejiang (Shengzhou) | 94 | UD | 0.09±0.10 | GFAAS | High (6.5) |
| 8 | Hu et al., 2019 | 2013 | South of Yangtze River Delta (Zhejiang) | 915 | 0.060±0.08 | 0.08±0.07 | Pb: ICP-OES Cd: ICP-MS | High (6.5) |
| 9 | Lu et al., 2018 | 2013 | Hunan | 440 | 0.049 ±0.004 | 0.565 ±0.376 | AAS | High (6.0) |
| 10 | Li et al., 2018 | 2013 | Yangtze River Delta region (Ningbo) | Rural: 10 | 0.027 ±0.034 | 0.071 ±0.061 | ICP-MS | Medium (5.5) |
| | | | | Industrial: 10 | 0.004 ±0.000 | 0.132 ±0.043 | | |
| 11 | Zeng et al., 2015 | 2013 | Hunan | 28 | 0.022 ±0.021 | 0.312 ±0.434 | GFAAS | High (6.0) |
| 12 | Tang et al., 2021 | 2014 | Guangxi (Liujiang District, Southern part of Liuzhou) | 75 | NR | 0.16±0.22 | ICP-MS | High (6.5) |
| 13 | Zheng et al., 2020 | 2014 | Pearl River Delta | 879 | 0.27±0.59 | 0.17±0.20 | Pb: FAAS Cd: GFAAS | High (6.5) |
| 14 | Huang et al., 2018 | 2014–2015 | Southeast China (Zhejiang) | 32 | 0.18±0.08 | 0.21±0.07 | Pb: ICP-OES Cd: ICP-MS | High (6.5) |
| 15 | Gu et al., 2019 | 2015 | Guangxi (Nanning and Laibin) | 246 | 0.042 ±0.020 | 0.182 ±0.171 | ICP-MS | High (6.5) |
| 16 | Mu et al., 2019 | 2015 | 19 provinces | 113 | 0.036 ±0.021 | 0.087 ±0.174 | ICP-MS | High (6.5) |
| | | | South/ Yangtze River Delta /West | 574 | 0.036 ±0.017 | 0.199 ±0.406 | | |
| 17 | Ma et al., 2017 | 2015 | Guangdong | 48 | 0.0274 ±0.0202 | 0.231 ±0.222 | ICP-MS | High (6.0) |
| 18 | Chen et al., 2018 | 2016 | Hunan (Xiangtan) | 200 | NR | 0.69±0.60 | ICP-MS | High (6.5) |
| 19 | He et al., 2019 | 2016 | Zhejiang (Wenling) | 169 | UD | 0.117 ±0.189 | GFAAS | High (6.5) |
| 20 | Wang et al., 2021 | 2016 | Guangdong (Shaoguan) | 570 | 0.19±0.092 | 0.62±0.94 | Pb: FAAS Cd: GFAAS | High (6.5) |
| 21 | Ren et al., 2021 | 2017 | Northern part of Zhejiang province | 120 | 0.04 ±0.05 | 0.09±0.07 | ICP-MS | High (6.0) |
| 22 | Zhang et al., 2020 | 2017 | Central part of Hunan | 135 | 0.145 ±0.328 | 0.283 ±0.330 | ICP-MS | High (6.5) |
| 23 | Guo et al., 2020 | 2018 | Centre of Zhejiang (Jin-Qu Basin) | 86 | 0.148 ±0.094 | 0.163 ±0.206 | ICP-MS | High (7.0) |

(*Continued*)

**Table 1.** (Continued)

| No. | Study | Year(s) of sampling | Area | Sample size | Level (mg/kg dry weight), mean±SD | | Assay method | Quality (Combie points) |
| --- | --- | --- | --- | --- | --- | --- | --- | --- |
| | | | | | Pb | Cd | | |
| 24 | Liu et al., 2020 | 2018 | Pearl River Delta (Zhuhai) | 70 | NR | 0.12±0.08 | ICP-MS | High (6.0) |
| 25 | Lu et al., 2021 | 2018 | Southwest of Fujian (Longyang) | 332 | 0.072 ±0.085 | 0.064 ±0.075 | ICP-MS | High (7.0) |
| 26 | Du et al., 2018 | NR | Hunan (Southern part of Changsha) | 27 | 0.031 ±0.023 | 0.291 ±0.295 | ICP-MS | Medium (5.0) |
| 27 | Lian et al., 2019 | NR | Shenyang | 41 | 0.26±0.026 | 0.14±0.016 | GFAAS | Medium (5.5) |
| 28 | Yu et al., 2019 | NR | Zhejiang (Nanxun, Shengzhou, Wenling) | Nanxun: 100 | NR | 0.011 ±0.015 | GFAAS | Medium (5.0) |
| | | | | Shengzhou: 94 | NR | 0.09±0.10 | | |
| | | | | Wenling: 96 | NR | 0.072 ±0.105 | | |
| 29 | Zhang et al., 2018 | NR | Guangdong (Sihui) | 31 | 2.05±4.67 | NR | ICP-OES | Medium (5.5) |
| 30 | Zhao et al., 2015 | NR | Zhejiang (Nanxun) | 100 | UD | 0.011 ±0.015 | GFAAS | Medium (5.5) |

AAS, atomic absorption spectrometry; FAAS, flame atomic absorption spectrometry; GFAAS, graphite furnace atomic absorption spectrometry; ICP, inductively coupled plasma; MS, mass spectrometry; NR, not reported; OES, optical emission spectroscopy; UD, undetectable (below the detection limit).

studies), atomic absorption spectrometry (AAS, 11 studies), and Cd were determined by inductively coupled plasma-mass spectrometry (15 studies), atomic absorption spectrometry (14 studies).

## Assessment of study quality

All studies in the review were judged to be of high or medium quality according to the Combie evaluation tool. The average score was 6.2 points, with 75.9% of the included studies scoring greater than 5.5 points (Table 1).

## Meta-analysis of concentrations of Pb and Cd

Of the 30 studies, four were excluded for the meta-analysis of Pb because concentrations were below the limit of detection in three studies [7, 38, 39], while the SD of concentrations in a fourth study [40] was 0.000. In the remaining studies, the pooled concentration of Pb (mg/kg) across several major rice-producing areas in China was 0.10 (95% CI 0.08–0.11; $I^2$ = 99.9%, P < 0.001; Fig 2). The pooled concentration of Cd (mg/kg) was 0.16 (95% CI 0.14–0.18; $I^2$ = 99.4%, P < 0.001; Fig 3).

Although some individual studies in our review reported levels of Pb or Cd in rice that exceeded the standard limit in China (0.2 mg/kg), the meta-analysis of pooled data demonstrated that the level of each metal was below this limit.

## Publication bias and sensitivity analysis

Egger's test suggested no significant risk of publication bias among studies measuring Pb (P = 0.712, Fig 4A), whereas it suggested significant risk among studies measuring Cd (P = 0.005, Fig 4B).

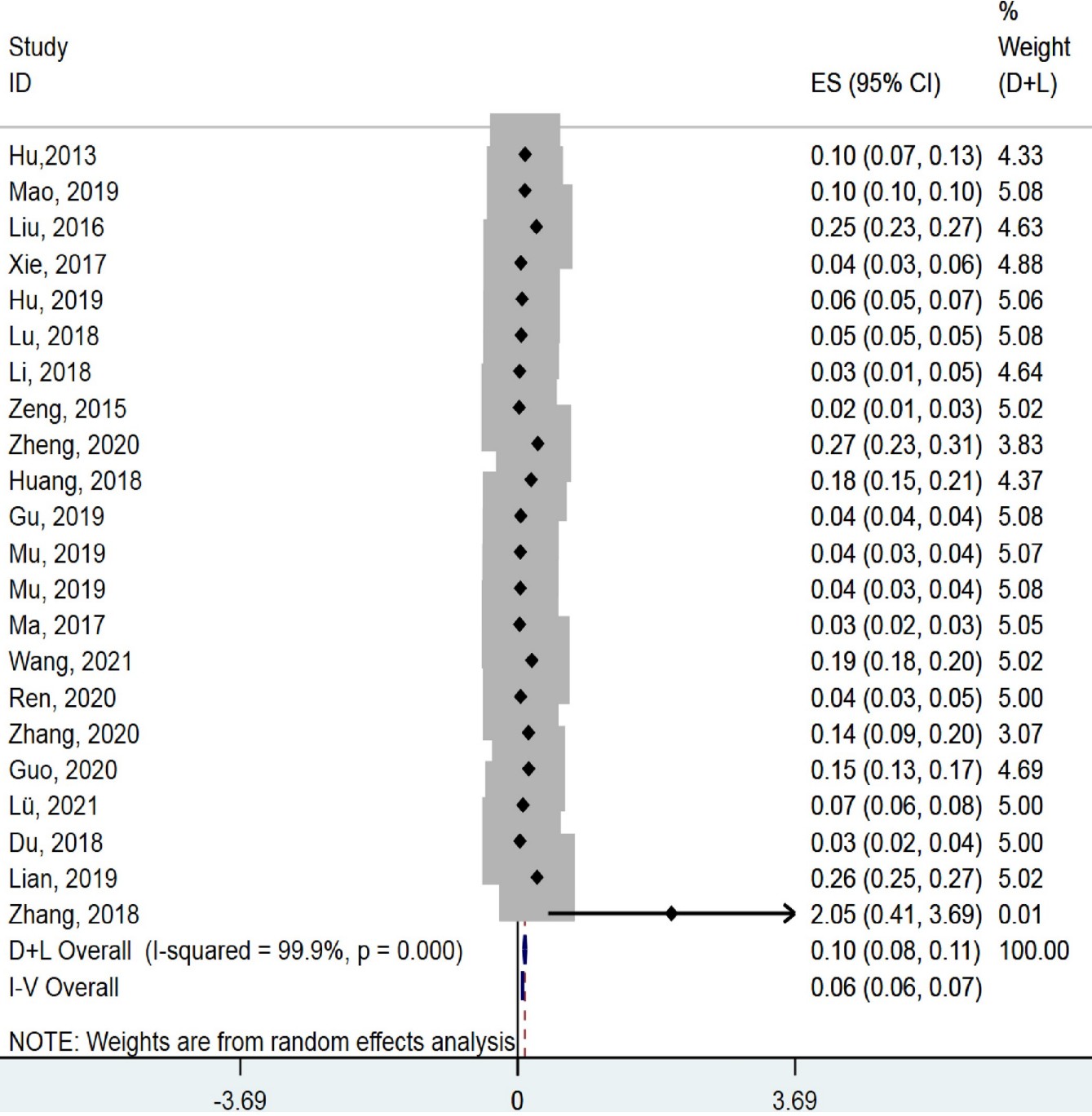

**Fig 2. Meta-analysis of Pb concentrations in rice.**

Sensitivity analysis was performed by repeating the meta-analysis after omitting each study one by one and examining whether the results changed substantially. Deletion of each one of the studies did not substantially alter the pooled concentrations of Pb or Cd (S1 Fig).

## Meta-regression analysis

Both uni- and multivariate meta-regressions were conducted with the following covariates: years of sampling, area, assay method, sample size and quality score. Univariate meta-

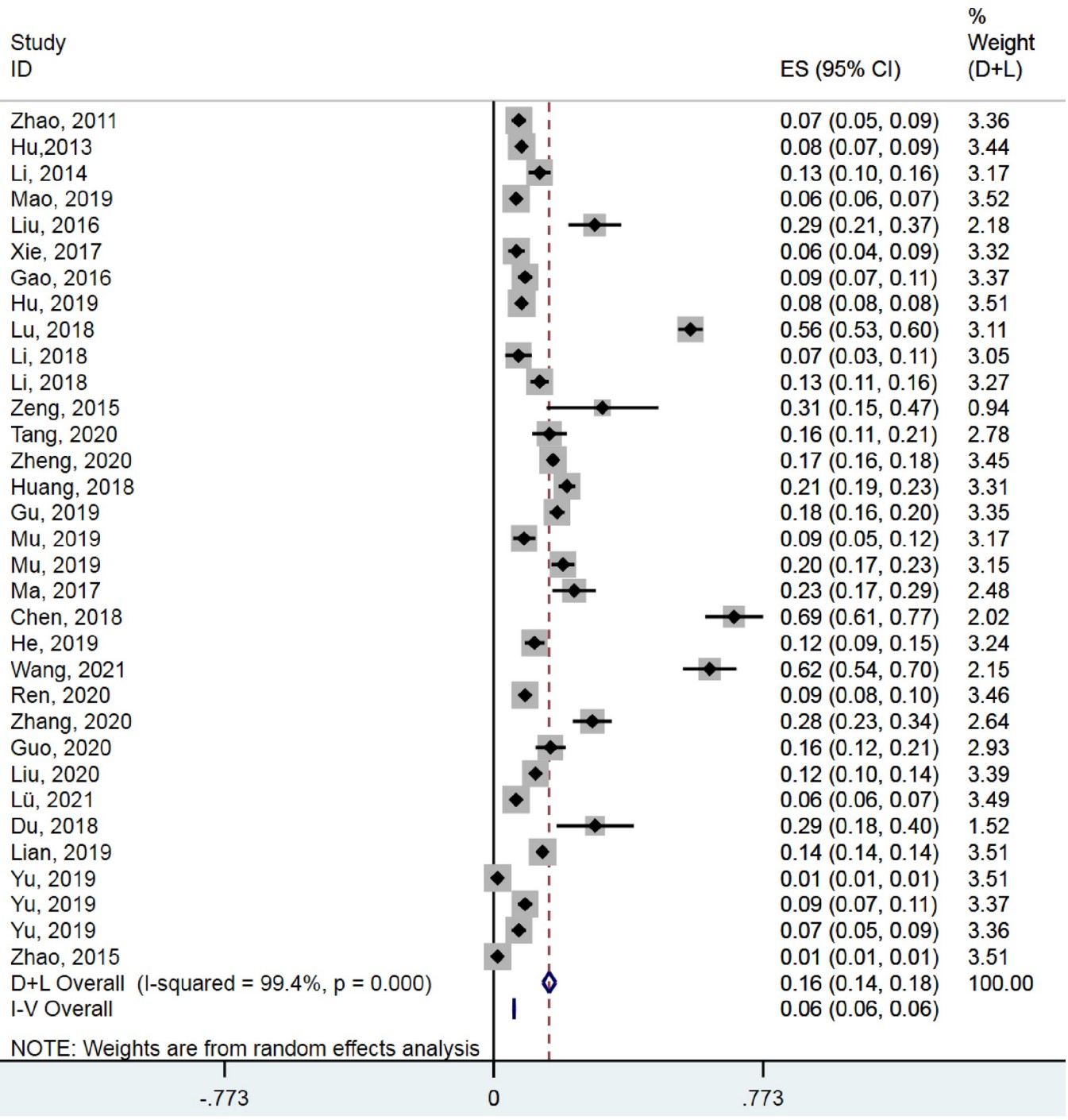

**Fig 3. Meta-analysis of Cd concentrations in rice.**

regression for Pb showed that years of sampling, area, assay method, sample size and quality score did not affect outcomes (Table 2). Nevertheless, assay method could explain 16.03% of heterogeneity (adjusted $R^2$ = 16.03%, P = 0.046). None of the factors tested substantially affected multivariate meta-regression (Table 3).

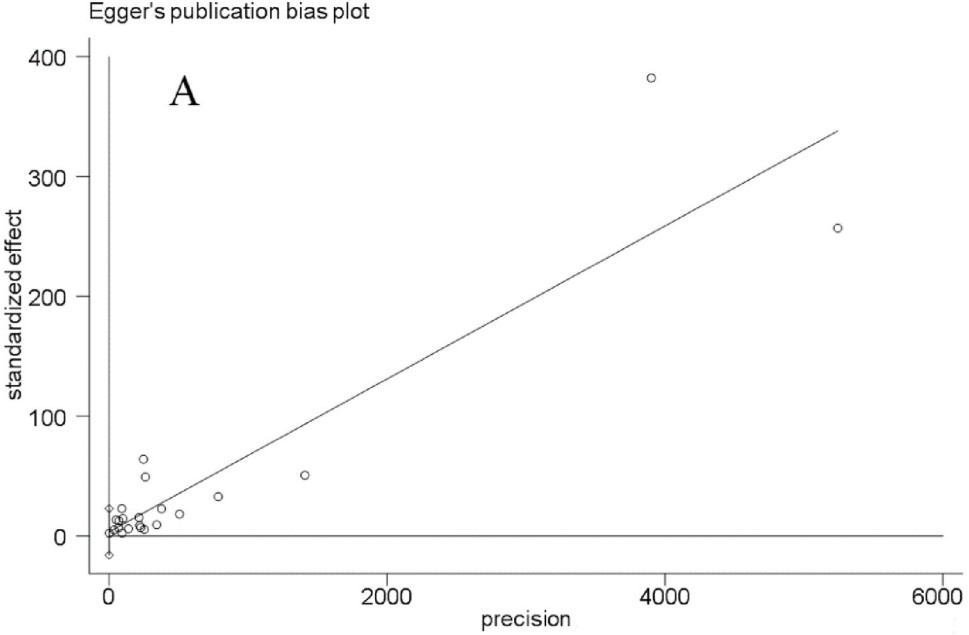

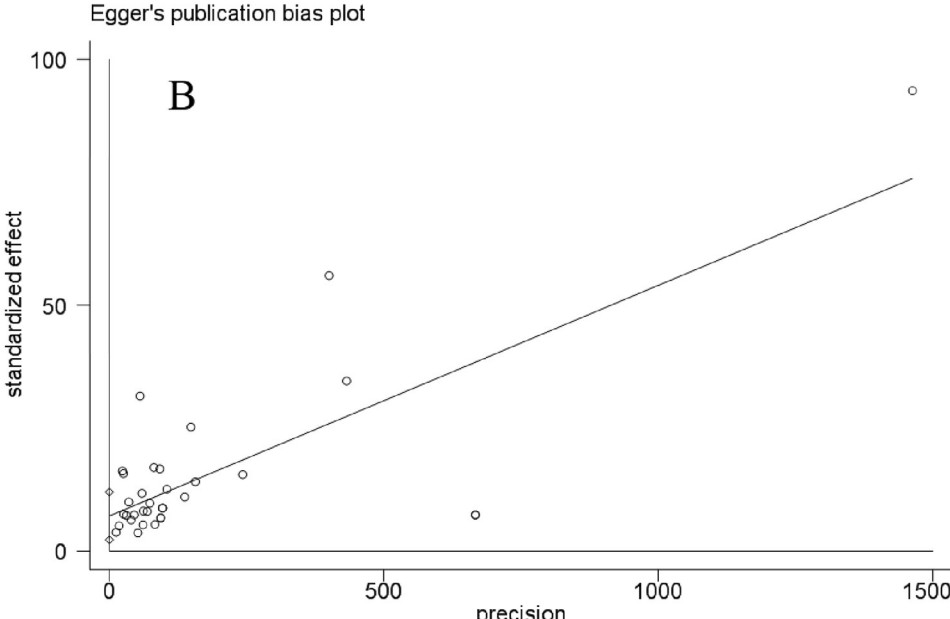

**Fig 4.** Egger's test to assess risk of publication bias among studies measuring (A) Pb or (B) Cd in rice samples.

Univariate meta-regression for Cd identified the following characteristics as affecting outcomes: northeast vs central China (adjusted $R^2$ = 47.81%, P = 0.040), eastern vs central China (adjusted $R^2$ = 47.81%, P<0.001), southern vs central China (adjusted $R^2$ = 47.81%, P = 0.007), central vs non-central China (adjusted $R^2$ = 43.90%, P<0.001), and sample size (adjusted $R^2$ = 15.56%, P = 0.016; Table 4). In contrast, years of sampling, assay method and quality score did not affect outcomes. Multivariate meta-regression showed that years of sampling, central vs non-central China, assay method, sample size and quality score were able to explain 41.86% of

**Table 2. Univariate meta-regression for Pb.**

| Covariate | Coefficient | 95% confidence interval | Adjusted $R^2$ | P |
|---|---|---|---|---|
| Years of sampling | 0.0065976 | -0.0196145 to 0.0328096 | -4.36% | 0.602 |
| Area of China | | | | |
| E vs N | -0.1681435 | -0.3993931 to 0.0631061 | 2.31% | 0.141 |
| C vs N | -0.161892 | -0.3967119 to 0.0729279 | 2.31% | 0.161 |
| S vs N | -0.1370138 | -0.3715523 to 0.0975248 | 2.31% | 0.231 |
| N vs non-N | 0.1567184 | -0.045606 to -0.045606 | 14.13% | 0.120 |
| Assay method | | | | |
| ICP-MS vs AAS | -0.0842295 | -0.1668027 to -0.0016563 | 16.03% | 0.046 |
| ICP-OES vs AAS | -0.0201361 | -0.1604507 to 0.1201785 | 16.03% | 0.767 |
| Sample size | 0.0000177 | -0.091281 to 0.0913164 | -5.42% | 1.000 |
| Quality score | -0.0134979 | -0.1359797 to 0.1089838 | -5.34% | 0.821 |

Regions of China were classified as follows: E, eastern (Zhejiang, Jiangsu, Shanghai); N, northeast (Liaoning); C, central (Hubei, Hunan, Jiangxi); S, southern (Guangxi, Guangdong, Fujian).

AAS, atomic absorption spectrometry; ICP, inductively coupled plasma; MS, mass spectrometry; OES, optical emission spectrometry.

heterogeneity (Table 5). The P value for the difference between central and non-central China was 0.002.

Meta-analysis showed high heterogeneity for Pb (99.9%) and Cd (99.4%). Uni- and multivariate meta-regression associated the high heterogeneity for Cd to different study areas in China.

## Subgroup analysis

Meta-analysis was repeated for specific subgroups defined in terms of years of sampling, area, assay method, sample size and quality score. Pooled concentrations of Pb (mg/kg) were as follows for different years of sampling (Table 6, Fig 5A): 2009–2011, 0.10 (95%CI 0.10, 0.10); 2012–2013, 0.07 (95%CI 0.05, 0.10); 2014–2015, 0.07 (95%CI 0.05, 0.08); 2016, 0.19 (95%CI 0.18, 0.20); 2017, 0.09 (95%CI -0.01, 0.19); and 2018, 0.11 (95%CI 0.03, 0.18).

Pooled concentrations of Cd (mg/kg) were as follows for different years of sampling (Table 7, Fig 5B): 2006, 0.07 (95%CI 0.05, 0.09); 2009–2011, 0.09 (95%CI 0.06, 0.11); 2012–2013, 0.19 (95%CI 0.11, 0.28); 2014–2015, 0.18 (95%CI 0.15, 0.20); 2016, 0.47 (95%CI 0.06, 0.89); 2017, 0.18 (95%CI 0.00, 0.37); 2018, 0.11 (95%CI 0.06, 0.16).

Regardless of years of sampling, levels of Pb were below the limit defined by China as safe. In contrast, the level of Cd exceeded the standard limit in 2016, but not in other years.

Pooled concentrations of Pb (mg/kg) were 0.26 (95%CI 0.25, 0.27) for northeast China, but 0.10 (95%CI 0.08, 0.12) across all other regions (Tables 6 and 8). Pooled concentrations of Cd

**Table 3. Multivariate meta-regression for Pb.**

| Covariate | Coefficient | 95% confidence interval | Adjusted $R^2$ | P |
|---|---|---|---|---|
| Years of sampling | 0.0186612 | -0.0203312 to 0.0576536 | -3.89% | 0.307 |
| Assay method | | | | |
| ICP-MS vs AAS | -0.1118847 | -0.2457248 to 0.0219555 | | 0.091 |
| ICP-OES vs AAS | -0.036351 | -0.1967322 to 0.1240303 | | 0.620 |
| Sample size | -0.0305863 | -0.1400625 to 0.0788899 | | 0.543 |
| Quality score | 0.0229515 | -0.198218 to 0.2441209 | | 0.820 |

Assay methods are defined in Table 2.

**Table 4. Univariate meta-regression for Cd.**

| Covariate | Coefficient | 95% confidence interval | Adjusted R² | P |
|---|---|---|---|---|
| Years of sampling | 0.0152284 | -0.0275551 to 0.0580119 | -2.00% | 0.470 |
| Area of China | | | | |
| N vs C | -0.2968999 | -0.5788897 to -0.0149101 | 47.81% | 0.040 |
| E vs C | -0.3322005 | -0.4702123 to -0.1941887 | 47.81% | 0.000 |
| S vs C | -0.2211105 | -0.3775602 to -0.0646608 | 47.81% | 0.007 |
| C vs non-C | 0.2980667 | 0.1612039 to 0.4349295 | 43.90% | 0.000 |
| Assay method | -0.0071547 | -0.1240696 to 0.1097602 | -3.38% | 0.901 |
| Sample size | 0.1437373 | 0.0285398 to 0.2589348 | 15.56% | 0.016 |
| Quality score | 0.1109727 | -0.0133534 to 0.2352988 | 7.26% | 0.078 |

Abbreviations for regions of China are defined in Table 2.

**Table 5. Multivariate meta-regression for Cd.**

| Covariate | Coefficient | 95% confidence interval | Adjusted R² | P |
|---|---|---|---|---|
| Years of sampling | 0.0092293 | -0.0370372 to 0.0554958 | 41.86% | 0.679 |
| Area: central vs non-central | 0.2869248 | 0.1182071 to 0.4556425 | | 0.002 |
| Assay method | 0.0520104 | -0.0969596 to 0.2009805 | | 0.471 |
| Sample size | 0.0768156 | -0.0605715 to 0.2142028 | | 0.254 |
| Quality score | 0.024864 | -0.1877351 to 0.2374632 | | 0.808 |

**Table 6. Subgroup analysis of Pb concentrations in rice.**

| Stratifying variable | Subgroup | No. of studies | Sample size | Concentration, mg/kg (95%CI) | P | I² (%) |
|---|---|---|---|---|---|---|
| Years of sampling | 2009–2011 | 2 | 229 | 0.10 (0.10, 0.10) | 0.891 | 0.0 |
| | 2012–2013 | 6 | 1604 | 0.07 (0.05, 0.10) | <0.001 | 98.8 |
| | 2014–2015 | 6 | 1892 | 0.07 (0.05, 0.08) | <0.001 | 98.1 |
| | 2016 | 1 | 570 | 0.19 (0.18, 0.20) | / | / |
| | 2017 | 2 | 255 | 0.09 (-0.01, 0.19) | <0.001 | 92.6 |
| | 2018 | 2 | 418 | 0.11 (0.03, 0.18) | <0.001 | 97.8 |
| | Not reported | 3 | 99 | 0.18 (-0.04, 0.40) | <0.001 | 99.9 |
| Area of China | Multiple areas | 4 | 889 | 0.04 (0.03, 0.05) | <0.001 | 85.2 |
| | Northeast | 1 | 41 | 0.26 (0.25, 0.27) | / | / |
| | Eastern | 6 | 1300 | 0.09 (0.06, 0.12) | <0.001 | 98.9 |
| | Central | 5 | 731 | 0.09 (0.06, 0.13) | <0.001 | 99.0 |
| | Southern | 6 | 2106 | 0.12 (0.06, 0.18) | <0.001 | 99.7 |
| | Northeast | 1 | 41 | 0.26 (0.25, 0.27) | / | / |
| | Non-Northeast | 17 | 4137 | 0.10 (0.08, 0.12) | <0.001 | 99.9 |
| Assay method | ICP-MS | 11 | 1828 | 0.06 (0.04, 0.09) | <0.001 | 99.9 |
| | ICP-OES | 3 | 978 | 0.13 (0.01, 0.25) | <0.001 | 97.3 |
| | AAS | 8 | 2261 | 0.15 (0.08, 0.22) | <0.001 | 99.8 |
| Sample size | ≤150 | 15 | 1111 | 0.10 (0.07, 0.13) | <0.001 | 99.7 |
| | >150 | 7 | 3956 | 0.09 (0.07, 0.10) | <0.001 | 99.7 |
| Quality score | High | 18 | 4958 | 0.10 (0.08, 0.11) | <0.001 | 99.9 |
| | Medium | 4 | 109 | 0.13 (-0.05, 0.30) | <0.001 | 99.8 |

Regions of China and assay methods are defined in Table 2.

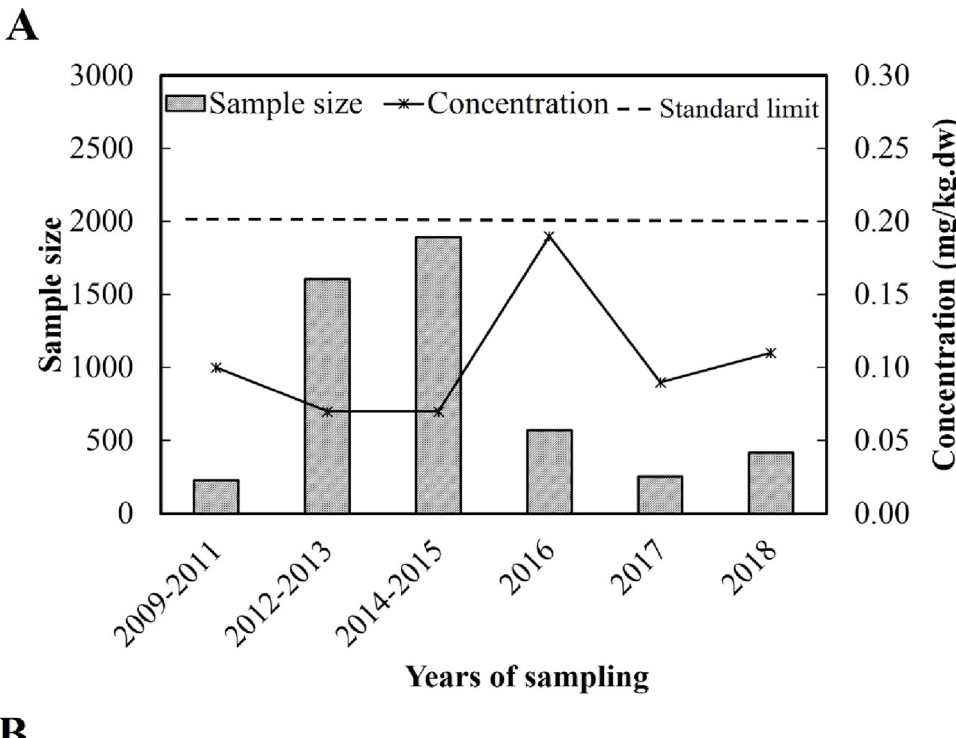

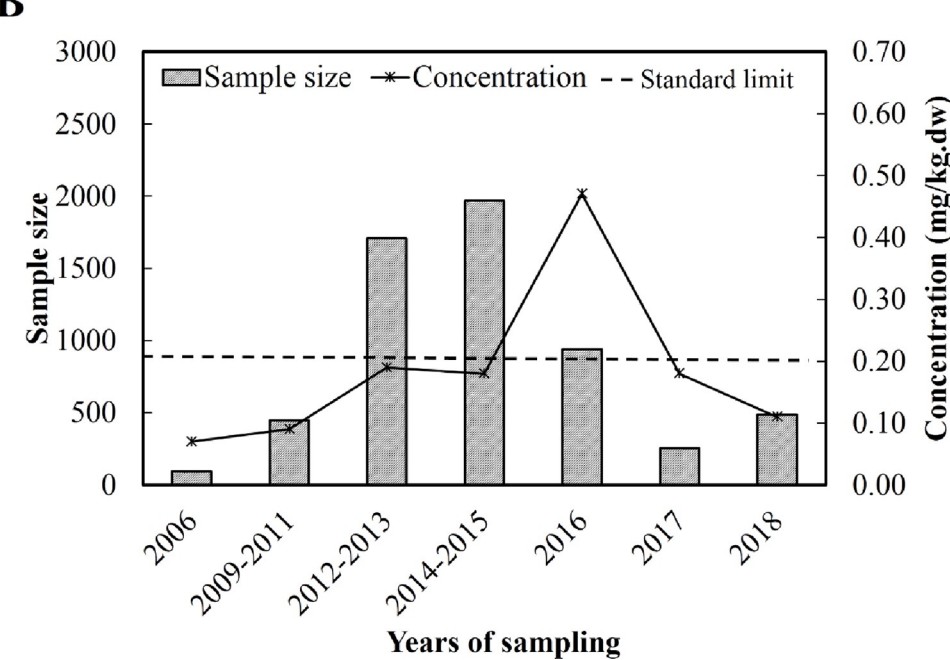

**Fig 5.** Pooled concentrations of (A) Pb and (B) Cd in different years of sampling. The dashed line indicates the safety limit defined by the Chinese government. dw, dry weight.

(kg/mg) were 0.43 (95%CI 0.27, 0.60) in central China, followed by 0.21 (95%CI 0.15, 0.27) in southern China, below 0.20 in other areas and 0.13 (95%CI 0.11, 0.15) across all non-central regions (Table 9). Heterogeneity was high for Cd measurements in central China ($I^2$ = 96.4%) as well as non-central regions (99.5%; Table 7).

**Table 7. Subgroup analysis of Cd concentrations in rice.**

| Stratifying variable | Subgroup | No. of studies | Sample size | Concentration 95%CI | P | $I^2$ (%) |
|---|---|---|---|---|---|---|
| Years of sampling | 2006 | 1 | 96 | 0.07 (0.05, 0.09) | / | / |
| | 2009–2011 | 3 | 448 | 0.09 (0.06, 0.11) | <0.001 | 91.0 |
| | 2012–2013 | 8 | 1708 | 0.19 (0.11, 0.28) | <0.001 | 99.1 |
| | 2014–2015 | 7 | 1967 | 0.18 (0.15, 0.20) | <0.001 | 86.1 |
| | 2016 | 3 | 939 | 0.47 (0.06, 0.89) | <0.001 | 99.3 |
| | 2017 | 2 | 255 | 0.18 (-0.00, 0.37) | <0.001 | 97.7 |
| | 2018 | 3 | 488 | 0.11 (0.06, 0.16) | <0.001 | 95.6 |
| | Not reported | 6 | 458 | 0.09 (0.04, 0.14) | <0.001 | 99.8 |
| Area of China | Multiple areas | 4 | 889 | 0.11 (0.06, 0.15) | <0.001 | 93.7 |
| | Northeast | 1 | 41 | 0.14 (0.14, 0.14) | / | / |
| | Eastern | 16 | 2379 | 0.10 (0.08, 0.12) | <0.001 | 99.4 |
| | Central | 5 | 830 | 0.43 (0.27, 0.60) | <0.001 | 96.4 |
| | Southern | 7 | 2220 | 0.21 (0.15, 0.27) | <0.001 | 98.6 |
| | Non-Central | 24 | 4640 | 0.13 (0.11, 0.15) | <0.001 | 99.5 |
| Assay method | ICP-MS | 17 | 3130 | 0.16 (0.14, 0.18) | <0.001 | 97.9 |
| | AAS | 16 | 3229 | 0.16 (0.13, 0.20) | <0.001 | 99.6 |
| Sample size | ≤150 | 23 | 1815 | 0.12 (0.10, 0.14) | <0.001 | 99.4 |
| | >150 | 10 | 4544 | 0.27 (0.21, 0.33) | <0.001 | 99.4 |
| Quality score | High | 24 | 5785 | 0.19 (0.17, 0.21) | <0.001 | 98.9 |
| | Medium | 9 | 574 | 0.09 (0.05, 0.13) | <0.001 | 99.7 |

Regions of China and assay methods are defined in Table 2.

Pooled concentrations of Pb (mg/kg) were as follows for different assay methods: ICP-MS, 0.06 (95%CI 0.04, 0.09); ICP-OES, 0.13 (95%CI 0.01, 0.25); and AAS, 0.15 (95%CI 0.08, 0.22) (Table 6). Pooled concentrations of Cd (mg/kg) were 0.16 (95%CI 0.14, 0.18) for ICP-MS and 0.16 (95%CI 0.13, 0.20) for AAS (Table 7).

Pooled concentrations of Pb (mg/kg) were 0.10 (95%CI 0.07, 0.13) among small studies (≤150 samples) and 0.09 (95%CI 0.07, 0.10) among large studies (>150 samples) (Table 6). Pooled concentrations of Cd (mg/kg) were 0.12 (95%CI 0.10, 0.14) among small studies and 0.27 (95%CI 0.21, 0.33) among large studies (Table 7).

Among studies measuring Pb, 18 were assigned to high quality and gave a pooled concentration of 0.10 (95%CI 0.08, 0.11) mg/kg. Four studies were assigned to medium quality and gave a pooled concentration of 0.13 (95%CI -0.05, 0.30) mg/kg (Table 6). Among studies measuring Cd, 24 were assigned to high quality and gave a pooled concentration of 0.19 (95%CI 0.17, 0.21) mg/kg. Nine studies were assigned to medium quality and gave a pooled concentration of 0.09 (95%CI 0.05, 0.13) mg/kg (Table 7).

**Table 8. Pooled concentrations of Pb in different areas of China.**

| Areas | Pb (mg/kg) |
|---|---|
| Northeast* | 0.26 (0.25, 0.27) |
| Eastern | 0.09 (0.06, 0.12) |
| Central | 0.09 (0.06, 0.13) |
| Southern | 0.12 (0.06, 0.18) |

Regions of China are defined as in Table 2. The * indicates exceed the standard limit.

**Table 9. Pooled concentrations of Cd in different areas of China.**

| Areas | Cd (mg/kg) |
|---|---|
| Northeast | 0.14 (0.14, 0.14) |
| Eastern | 0.10 (0.08, 0.12) |
| Central* | 0.43 (0.27, 0.60) |
| Southern* | 0.21 (0.15, 0.27) |

Regions of China are defined as in Table 2. The
* indicates exceed the standard limit.

**Table 10. THQ and total THQ of Pb and Cd due to rice consumption.**

| Group | Pb-THQ | Cd-THQ | Total THQ |
|---|---|---|---|
| Adults | 0.20 | 1.11 | 1.31 |
| Children | 0.17 | 0.97 | 1.14 |

THQ, target hazard quotient.

Our meta-analysis indicated more serious contamination of rice with Cd than with Pb. Contamination with Cd appears particularly severe in the central region of China (0.43 mg/kg), based primarily on pooled data from Hunan [4, 12, 41–44] but also some data from Jiangxi and Hubei [14]. Our findings are consistent with several studies reporting widespread soil contamination with Cd in Hunan, where some types of local rice are referred to as "cadmium rice" [12, 45, 46].

Although our studies sampled from all six of the major rice-producing regions in China, the sampling was concentrated in Zhejiang in the Yangtze River Delta and Guangdong in southern China. Given that levels of heavy metals in rice appear to vary geographically [24], we recommend that future studies focus on neglected rice-producing regions in China in order to provide a more comprehensive and accurate picture of heavy metal contamination.

## Health risk assessment

Our meta-analysis of the literature suggests a Pb THQ of 0.20 for adults and 0.17 for children (Table 10), both of which are below 1.0, indicating safe levels in rice. In contrast, the Cd THQ was 1.11 for adults and 0.97 for children, indicating a health concern for adults but not children. Combining the THQs for Pb and Cd led to a total THQ higher than 1 for adults and children. This suggests a serious health risk for children and adults.

## Conclusions

Our meta-analysis suggests that pooled Pb and Cd levels are within the limits specified by Chinese food safety standards. Nevertheless, the total target hazard quotient for both metals appears to exceed 1.0 for adults and children, suggesting that rice consumption poses a health risk and more should be done to control heavy metal pollution of soils in rice paddies in China.

## Supporting information

**S1 Fig. Sensitivity analysis.**
(DOCX)

## Author Contributions

**Conceptualization:** Xiaoshan Yang.

**Data curation:** Mingtian Tan, Lisha Shen.

**Formal analysis:** Xianliang Huang, Mingtian Tan.

**Funding acquisition:** Xianliang Huang, Xiaoshan Yang.

**Investigation:** Xianliang Huang.

**Project administration:** Hongyu Zhou.

**Resources:** Yanlei Wu, Guirong Feng.

**Supervision:** Bo Zhao, Shiqi Chen, Hongyu Zhou.

**Writing – original draft:** Xianliang Huang, Bo Zhao, Yanlei Wu.

**Writing – review & editing:** Xianliang Huang, Youming Xiong, En Zhang.

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
