## [Decision Letter · Decision Letter 0]

6 Sep 2022

PONE-D-22-16207Heavy metals in rice and risk to human health in China: a systematic review and meta-analysisPLOS ONE

Dear Dr. ZHOU,

Thank you for submitting your manuscript to PLOS ONE. After careful consideration, we feel that it has merit but does not fully meet PLOS ONE’s publication criteria as it currently stands. Therefore, we invite you to submit a revised version of the manuscript that addresses the points raised during the review process.

Kindly revise according to the reviewers comments.  Note from Staff Editor Hanna Landenmark: Please note that Reviewer 1 has submitted a review for the wrong manuscript, and thus this review should be disregarded. Please address only reviewer 2-4.

We look forward to receiving your revised manuscript.

Kind regards,

Sartaj Ahmad Bhat, Ph.D

Academic Editor

PLOS ONE

Journal Requirements:

"This work was supported by the National Key R&D Program of China and Chongqing Science and Technology Bureau."

5. We note that Figure 6 in your submission contain map image which may be copyrighted. All PLOS content is published under the Creative Commons Attribution License (CC BY 4.0), which means that the manuscript, images, and Supporting Information files will be freely available online, and any third party is permitted to access, download, copy, distribute, and use these materials in any way, even commercially, with proper attribution. For these reasons, we cannot publish previously copyrighted maps or satellite images created using proprietary data, such as Google software (Google Maps, Street View, and Earth). For more information, see our copyright guidelines: http://journals.plos.org/plosone/s/licenses-and-copyright.

a. You may seek permission from the original copyright holder of Figure 6 to publish the content specifically under the CC BY 4.0 license.  

Additional Editor Comments:

Note from Staff Editor Hanna Landenmark: Please note that Reviewer 1 has submitted a review for the wrong manuscript, and thus this review should be disregarded. Please address only reviewer 2-4.

Reviewers' comments:

Reviewer's Responses to Questions

**Comments to the Author**

1. Is the manuscript technically sound, and do the data support the conclusions?

Reviewer #1: Yes

Reviewer #2: Partly

Reviewer #3: No

Reviewer #4: Yes

2. Has the statistical analysis been performed appropriately and rigorously? 

Reviewer #1: No

Reviewer #2: Yes

Reviewer #3: Yes

Reviewer #4: Yes

3. Have the authors made all data underlying the findings in their manuscript fully available?

Reviewer #1: Yes

Reviewer #2: No

Reviewer #3: Yes

Reviewer #4: Yes

4. Is the manuscript presented in an intelligible fashion and written in standard English?

Reviewer #1: Yes

Reviewer #2: Yes

Reviewer #3: No

Reviewer #4: Yes

5. Review Comments to the Author

Reviewer #1: The presented manuscript entitled “Heavy Metal Concentrations in a Soil-Plant-Ruminant food Chain along a Terrestrial Soil Pollution Gradient: Health Risk Assessment” deals with the assessment of toxic metals concentrations in soil-plant-ruminant food chain along a terrestrial soil pollution gradient. In the present scenario, safe disposal of heavy metals from various sectors including domestic is one of the global challenges for sustainable development. This is interesting research that describes a hot topic. In this manuscript, a good effort has been done by the authors to explore the potential of metal in soil plant-ruminant food chain as well as terrestrial soil pollution gradient. In general, the topic is in line with the journal's scope. Overall, the manuscript has scientific merits and well structured. Hence, I recommend the manuscript for publication in Frontiers in Environmental Science. However, the following are the specific comments which need to be addressed:

Abstract is written good but could you provide more informative results/data so outcomes may reflect the novelty of the paper. The abstract must be carefully prepared to attract readers to read the full text and then cite it in their future publications. Also, more quantitative information needs to be provided in the abstract. Provide numerical data.

Define acronyms when they first appear; thereafter directly use them.

Use SI units; follow the correct format (e.g., mg kg-1). Check through the entire paper to make sure it is defined at the first use.

The researchers should revise the introduction section to include more information about the core topic, its importance and research direction. The authors should briefly discuss the purpose of the research and clearly mention their findings adopted in this study.

I have one important concern about the novelty of this work. The originality/novelty of the paper should be clearly stated in the manuscript.

There is no explanation about the importance of using the statistical analysis approach in the study

Discussion is better described, but the authors should state each citation to its specific discovery.

Clearly discuss what the previous studies that you are referring to are. What are the Research Gaps/Contributions?

Conclusion: What is the "take-away-message" of the paper? What is the novelty of the paper? Conclusion is not showing any significant findings. In your conclusions, please discuss the implications of your research. Conclusions must go deeper, it would be more interesting if the authors focus more on the significance of their findings regarding the importance of the interrelationship between the obtained results and sustainable development/cleaner production in the sector context, and the barriers to do it, what would be the consequences, in the real world, in changing the observed situation, what would be the ways, in the real world, to change/improve the observed situation.

The technical language of the manuscript is good. The few grammatically errors should be checked.

The Table legends, figure captions, and foot notes need improvement. All legends, captions, and foot notes should have enough description for a reader to understand the figure without having to refer back to the main text of the manuscript.

For citations and reference within the text, the author must follow guide for authors. The references must be also in the format of the journal.

Reviewer #2: Authors have worked rigorously for this review and meta analysis, but few major revisions are required before acceptance of this manuscript. In addition to the revisions in attachments following changes needs to be done:

- The final number of studies selected are very few for meta analysis and review. I think some parameters for selection should be modified for inclusion of more studies. Also, considering the amount of work done and number of studies published form China about Pb and Cd content in rice, the number of studies selected for this review are too low.

- I would also suggest to expand the time of study to 20 year period (2000-2020). It would add a temporal dimension to the study also as authors could deduce results about chnage in Pb and Cd accumulaiton in rice over 20 years of time. Thus interesting trends can emerge from the same geographical areas over 20 years time.

- I think addition of sources of these metals in the studies are very important and authors must add section for source apportionment of these two metals also, which should include the differecne sources mentioned by auhtors regarding these two metals.

Check the attachments for further comments.

Reviewer #3: Related with the manuscript entitled "Heavy metals in rice and risk to human health in China: a systematic review and meta-analysis", I have to say that it brings nothing new. The paper was not built properly , especially without a useful and critical discussion.

Reviewer #4: In general,l this is good risk assessment of Cd and Pb in the rice. Very much of public concern indeed.

You only focused on Cd and Pb, so please revise your title of paper.

However, in your calculation of THQ, is your C (concentrations of Cd and Pb) in wet weight basis?

Be in mind that this THQ should be based on the cooked rice with water (not dried rice or grains). So, you need to know the water content of rice that we cooked by knowing the conversion factor to the wet basis (moisture content).

Of course, some people prefer to put more water and some less water. This is sometimes debatable as well (do make the discussion as well).

The THQ of metals is based the direct consumption orally (via the months). I hope you can understand my point well.

Please check and confirm or re-calculate that THQ is based on the rice (with water during cooking), so, wet weight as I said.

Thank you

Good luck

6. PLOS authors have the option to publish the peer review history of their article (what does this mean?). If published, this will include your full peer review and any attached files.

Reviewer #1: No

Reviewer #2: No

Reviewer #3: No

Reviewer #4: No

---

## [Author Response · Author response to Decision Letter 0]

21 Oct 2022

Dear editors and reviewers:

Thank you for your letter and the comments concerning our manuscript entitled “Heavy metals in rice and risk to human health in China: a systematic review and meta-analysis” (ID: PONE-D-22-16207). Those comments are all valuable and very helpful for revising and improving our paper, as well as the important guiding significance to our researches. We have studied comments carefully and have made correction which we hope meet with approval. All suggestions and the reply to the comments are as follows:

Editor:

Response: The manuscript has been revised according to PLOS ONE's style requirements.

Response: I'm sorry I made a mistake and didn't explain it clearly. The grant numbers for these three awards are correct. In fact, both cstc2021jxjl130009 and cstc2018jscx-mszdX0122 are funded by Chongqing Science and Technology Bureau.

The ‘Funding Information’ has been revised to:

(1) Recipient: X.S.Y. Grant number: 2017YFC1602000, Funding Source: the National Key R&D Program of China.

(2) Recipient: X.L.H. Grant number: cstc2021jxjl130009, Funding Source: Chongqing Performance Incentive Guidance Special Project of Chongqing Science and Technology Bureau.

(3) Recipient: X.S.Y. Grant number: cstc2018jscx-mszdX0122, Funding Source: the Key demonstration project of Chongqing Technology Innovation and application demonstration project of Chongqing Science and Technology Bureau.

"This work was supported by the National Key R&D Program of China and Chongqing Science and Technology Bureau."

Response: We wish to change our statement to "The funders had no role in study design, data collection and analysis, decision to publish, or preparation of the manuscript. "

Response: We wish to change our Data Availability Statement to "All relevant data are within the manuscript and its Supporting Information files".

5. We note that Figure 6 in your submission contain map image which may be copyrighted. All PLOS content is published under the Creative Commons Attribution License (CC BY 4.0), which means that the manuscript, images, and Supporting Information files will be freely available online, and any third party is permitted to access, download, copy, distribute, and use these materials in any way, even commercially, with proper attribution. For these reasons, we cannot publish previously copyrighted maps or satellite images created using proprietary data, such as Google software (Google Maps, Street View, and Earth). For more information, see our copyright guidelines: http://journals.plos.org/plosone/s/licenses-and-copyright.

Response: We replaced Figure 6 with Tables 8 and 9. 

Response: Supporting information captions has been listed at the end of the manuscript.

Supporting information

S1 Fig. The sensitivity analysis of Pb (given named study was omitted).

S2 Fig. The sensitivity analysis of Cd (given named study was omitted).

Reviewer: 2

Authors have worked rigorously for this review and meta analysis, but few major revisions are required before acceptance of this manuscript. In addition to the revisions in attachments following changes needs to be done:

1. The final number of studies selected are very few for meta analysis and review. I think some parameters for selection should be modified for inclusion of more studies. Also, considering the amount of work done and number of studies published form China about Pb and Cd content in rice, the number of studies selected for this review are too low.

Response: Indeed, there are many research reports on the content of lead and cadmium in rice in China. However, lots of them have been excluded, the reasons are as follows: Firstly, some of them were sampled in the markets, these samples may be produced in China or other countries, and the specific origin is unknown. The objective of our research is to analyze the rice produced in China, so, rice samples collected from markets were excluded. Secondly, many comparative studies are conducted in mining areas or polluted areas, which are not suitable for rice planting, just for some experimental researches, so they are not included. Thirdly, some studies published in Chinese rather than English, which are excluded in order to ensure the quality of included articles. 

Although the number of studies included is not very large, they involved a total of 6390 rice samples from several major rice-producing areas in China, the meta analysis and review could be carried out and support our research conclusions.

2. I would also suggest to expand the time of study to 20 year period (2000-2020). It would add a temporal dimension to the study also as authors could deduce results about change in Pb and Cd accumulation in rice over 20 years of time. Thus interesting trends can emerge from the same geographical areas over 20 years time.

Response: We searched the literature from 2000 to 2010, which is much less than that from 2011 to 2021, and few can meet the inclusion criteria, may not be convincing in reflecting the trend of relevant regions. However, this is really a good suggestion. In future research, we can try to search the data information of other heavy metals between 2000 and 2020 to find if there are some interesting trends.

3. I think addition of sources of these metals in the studies are very important and authors must add section for source apportionment of these two metals also, which should include the differecne sources mentioned by auhtors regarding these two metals.

Check the attachments for further comments.

Response: We have revised to “Contamination of heavy metals is mainly caused by natural origination and anthropogenic activities, of which the latter one (include industries of mining, fertilizers, and pesticides) made predominant contribution, have led to the continuing accumulation of toxic heavy metals in the soil of rice paddies, from which the metals can enter rice.”

The followings are raised by reviewer 2 in the attachments.

1. There are several other sources for heavy metal accumulation which must be mentioned here.

Response: We have revised to “Contamination of heavy metals is mainly caused by natural origination and anthropogenic activities, of which the latter one (include industries of mining, fertilizers, and pesticides) made predominant contribution, have led to the continuing accumulation of toxic heavy metals in the soil of rice paddies, from which the metals can enter rice.”

2. Authors must mention the agency for whose limit is quoted. (Line 46)

Response: The agency has been added and marked in the manuscript.

3. Objectives of the study must be mentioned in clear point wise manner.

Response: Objectives of the study has been revised to “Therefore we aimed to 1) investigate, even via a meta-analysis of the existing literature, the presence of Pb and Cd in rice from many areas in China; and 2) assess the potential human health risks associated with long-term exposure.”

4. How were the ranges converted to SD? Explain.

Response: When the sample size between 25 and 70, Range/4 is the best estimator for the standard deviation.

The explanation has been added and marked in the manuscript.

5. Brief explanation of Combie evaluation tool must be given.

Response: Included studies were graded in 7 aspects according to the Combie evaluation tool which is as follows: the study design was scientific and rigorous; the data collection method was reasonable; the response rate of participants was reported; the total representativeness of samples were favorable; the research objective and methods were reasonable; the power of the test was reported; the statistical method was correct. “Yes”, “no” and “have no idea” were respectively utilized to evaluate each item, which was successively given 1 point, 0 points, and 0.5 points. The total score was 7.0 points (6.0～7.0 points, 4.0～5.5 points, and 0～4.0 points were considered to high, medium and low quality respectively)

The explanation has been added and marked in the manuscript.

6. In my assumption the number of studies selected are very few.

Response: In order to ensure the quality of the included studies to meet the needs of our research objectives, most of the articles were excluded. Although the number of studies included is not very large, they involved a total of 6390 rice samples from several major rice-producing areas in China, the meta analysis and review could be carried out and support our research conclusions.

7. Why only the specific 10 year period was chosen and not more? If authors selected the 20 year period of 2000-2020 it would give a very nice temporal dimension to the above study.

Response: We searched the literature from 2000 to 2010, which is much less than that from 2011 to 2021, and few can meet the inclusion criteria, may not be convincing in reflecting the trend of relevant regions. However, this is really a good suggestion. In future research, we can try to search the data information of other heavy metals between 2000 and 2020 to find if there are some interesting trends.

Reviewer: 3

Related with the manuscript entitled "Heavy metals in rice and risk to human health in China: a systematic review and meta-analysis", I have to say that it brings nothing new. The paper was not built properly , especially without a useful and critical discussion.

Response: There were few reports on the analysis of lead and cadmium content in rice in China by means of systematic review and meta analysis. In addition, we collected articles published from January 2011 through October 2021 in English for analysis. It is rare to use so much data for analysis. Our findings suggest that rice consumption poses a health risk and more should be done to control heavy metal pollution of soils in rice paddies in China.

We collected 10 years of literatures, analyzed a large number of data, and reached the above conclusions, which is rarely reported before.

Reviewer: 4

1. In general,l this is good risk assessment of Cd and Pb in the rice. Very much of public concern indeed.

You only focused on Cd and Pb, so please revise your title of paper.

Response: We have revised the title of paper to “The lead and cadmium content in rice and risk to human health in China: a systematic review and meta-analysis”.

2. However, in your calculation of THQ, is your C (concentrations of Cd and Pb) in wet weight basis?

Be in mind that this THQ should be based on the cooked rice with water (not dried rice or grains). So, you need to know the water content of rice that we cooked by knowing the conversion factor to the wet basis (moisture content).

Of course, some people prefer to put more water and some less water. This is sometimes debatable as well (do make the discussion as well).

The THQ of metals is based the direct consumption orally (via the months). I hope you can understand my point well.

Please check and confirm or re-calculate that THQ is based on the rice (with water during cooking), so, wet weight as I said.

Response: Thank you for your reminder! The concentrations of Cd and Pb reported in the articles included in the data analysis are dry weight basis, and C in our study are dry weight basis. The THQ of metals is based the direct consumption orally, and FIR (the average daily rice intake in kg person−1 day–1) used in our calculation is also measured by dry weight, not after cooking.

---

## [Decision Letter · Decision Letter 1]

22 Nov 2022

The lead and cadmium content in rice and risk to human health in China: a systematic review and meta-analysis

PONE-D-22-16207R1

Dear Dr. Zhou,

We’re pleased to inform you that your manuscript has been judged scientifically suitable for publication and will be formally accepted for publication once it meets all outstanding technical requirements.

Kind regards,

Sartaj Ahmad Bhat, Ph.D

Academic Editor

PLOS ONE

Additional Editor Comments (optional):

Reviewers' comments:

Reviewer's Responses to Questions

**Comments to the Author**

1. If the authors have adequately addressed your comments raised in a previous round of review and you feel that this manuscript is now acceptable for publication, you may indicate that here to bypass the “Comments to the Author” section, enter your conflict of interest statement in the “Confidential to Editor” section, and submit your "Accept" recommendation.

Reviewer #1: All comments have been addressed

Reviewer #3: All comments have been addressed

2. Is the manuscript technically sound, and do the data support the conclusions?

Reviewer #1: Yes

Reviewer #3: Yes

3. Has the statistical analysis been performed appropriately and rigorously? 

Reviewer #1: Yes

Reviewer #3: Yes

4. Have the authors made all data underlying the findings in their manuscript fully available?

Reviewer #1: Yes

Reviewer #3: Yes

5. Is the manuscript presented in an intelligible fashion and written in standard English?

Reviewer #1: Yes

Reviewer #3: Yes

6. Review Comments to the Author

Reviewer #1: (No Response)

Reviewer #3: (No Response)

7. PLOS authors have the option to publish the peer review history of their article (what does this mean?). If published, this will include your full peer review and any attached files.

Reviewer #1: No

Reviewer #3: No

---

## [Editor Report · Acceptance letter]

7 Dec 2022

PONE-D-22-16207R1 

The lead and cadmium content in rice and risk to human health in China: a systematic review and meta-analysis 

Dear Dr. Zhou:

I'm pleased to inform you that your manuscript has been deemed suitable for publication in PLOS ONE. Congratulations! Your manuscript is now with our production department. 

Kind regards, 

on behalf of

Dr. Sartaj Ahmad Bhat 

Academic Editor

PLOS ONE